# Can Modern Molecular Modeling Methods Help Find the Area of Potential Vulnerability of Flaviviruses?

**DOI:** 10.3390/ijms23147721

**Published:** 2022-07-13

**Authors:** Daniil V. Shanshin, Sophia S. Borisevich, Alexander A. Bondar, Yuri B. Porozov, Elena A. Rukhlova, Elena V. Protopopova, Nikita D. Ushkalenko, Valery B. Loktev, Andrei I. Chapoval, Alexander A. Ilyichev, Dmitriy N. Shcherbakov

**Affiliations:** 1State Research Center of Virology and Biotechnology VECTOR, Rospotrebnadzor, 630559 Koltsovo, Russia; ruhlovalena@mail.ru (E.A.R.); protopopova_ev@vector.nsc.ru (E.V.P.); ushkalenko@yahoo.com (N.D.U.); valeryloktev@gmail.com (V.B.L.); ilyichev@vector.nsc.ru (A.A.I.); dnshcherbakov@gmail.com (D.N.S.); 2Laboratory of Physical Chemistry, Ufa Institute of Chemistry, Ufa Federal Research Center Russian Academy of Science, 450054 Ufa, Russia; 3Russian-American Anti-Cancer Center, Altai State University, 656049 Barnaul, Russia; andreichapoval@gmail.com; 4Institute of Chemical Biology and Fundamental Medicine, 630090 Novosibirsk, Russia; alex.bondar@mail.ru; 5The Center of Bio- and Chemoinformatics, I.M. Sechenov First Moscow State Medical University, 119435 Moscow, Russia; yuri.porozov@gmail.com; 6Department of Computational Biology, Sirius University of Science and Technology, Olympic Ave 1, 354340 Sochi, Russia

**Keywords:** flavivirus, TBEV, WNV, ZIKV, DENV, monoclonal antibody, recombinant protein, ELISA, molecular protein docking, molecular protein dynamic

## Abstract

Flaviviruses are single-stranded RNA viruses that have emerged in recent decades and infect up to 400 million people annually, causing a variety of potentially severe pathophysiological processes including hepatitis, encephalitis, hemorrhagic fever, tissues and capillaries damage. The *Flaviviridae* family is represented by four genera comprising 89 known virus species. There are no effective therapies available against many pathogenic flaviviruses. One of the promising strategies for flavivirus infections prevention and therapy is the use of neutralizing antibodies (NAb) that can disable the virus particles from infecting the host cells. The envelope protein (E protein) of flaviviruses is a three-domain structure that mediates the fusion of viral and host membranes delivering the infectious material. We previously developed and characterized 10H10 mAb which interacts with the E protein of the tick-borne encephalitis virus (TBEV) and many other flaviviruses’ E proteins. The aim of this work was to analyze the structure of E protein binding sites recognized by the 10H10 antibody, which is reactive with different flavivirus species. Here, we present experimental data and 3D modeling indicating that the 10H10 antibody recognizes the amino acid sequence between the two cysteines C92-C116 of the fusion loop (FL) region of flaviviruses’ E proteins. Overall, our results indicate that the antibody-antigen complex can form a rigid or dynamic structure that provides antibody cross reactivity and efficient interaction with the fusion loop of E protein.

## 1. Introduction

In recent years, there has been a constant increase in infectious diseases caused by viruses from the *Flaviviridae* family [1,2]. The *Flaviviridae* family is represented by four genera comprising of 89 known virus species. All viruses of the *Flaviviridae* family share the genome structure and the amino acid homology of major structural proteins [3,4]. Diseases induced by the viruses of the *Flaviviridae* family such as dengue virus (DENV), West Nile virus (WNV), tick-borne encephalitis virus (TBEV), Japanese encephalitis virus (JEV) and Zika virus (ZIKV) are among the most life-threatening emerging infections that lead to more than 500,000 annual hospitalizations with a mortality rate of about 5%, caused by hemorrhagic fever, shock syndrome, and tissue and capillaries damage [5,6,7,8,9].

There are effective inactivated vaccines against JEV, TBEV, and an attenuated vaccine against yellow fever virus (YFV) [10,11,12]. However, the development of vaccines against DENV, WNV and ZIKV is complicated probably due to the antibody-dependent enhancement (ADE) of infections [13,14,15]. An alternative to the vaccine may be serum therapy. However, obtaining specific immunoglobulins from donors’ blood is limited by material availability and possible viral contamination. In addition, possible antibody-dependent disease severity enhancement also significantly limits the use of serum therapy [13,14,15].

In some cases, monoclonal antibodies (mAb) can be used for flavivirus-infection therapy. Currently, several monoclonal neutralizing antibodies (NAb) against flaviviruses are available [4]. The targets recognized by neutralizing antibodies are usually the membrane protein precursor [16], nonstructural protein 1 (NS1) [17], and envelope protein (E protein) [18]. The E protein is involved in the binding of virus particles to cell receptors that mediate the penetration of the viruses to host cells. This feature makes the E protein an attractive target for NAb development. The extracellular part of the E protein consists of the following three domains: the central ß-barrel domain I (DI), which is connected to the dimerization domain II (DII), and the binding domain to the Ig-like receptor III (DIII) [4]. DII contains a hydrophobic sequence, called the fusion loop (FL), which is conserved among most *Flavivirus* species [4]. FL is immersed in the host cell membrane during pH-induced conformational changes. This process results in the fusion of viral and host cell membranes [4].

Neutralizing antibodies against the E protein can inhibit several processes during infection, including binding to cellular receptors, blocking conformational changes, and affecting FL function, thereby preventing viral particle entry to the host cells [4]. The FL region of the E protein is highly conserved for all flaviviruses and may be a potential target blocking virus entry to the host cells. Several antibodies against this region have been developed and described, including 3G9 [19], POWV-56 [20], 2A10G6 [21], E53 [22] and 758P6B10 [23]. However, the above antibodies interacting with the FL region have limited neutralization efficacy due to FL conformation in accessing mature virions. The FL region becomes accessible for antibodies only during the fusion between the viral and host cell membrane [4]. The ADI-15878 and VRC34.01 antibodies recognize the FL region of E proteins in Ebola and HIV-1 viruses and are capable of blocking virus entry to the host cells with high efficiency [24,25,26].

Previously, we determined and described mouse monoclonal non-neutralizing Ab 10H10 that interacts with TBEV and many other flaviviruses [27]. The data suggest that 10H10 mAb recognizes the FL region of the E protein [28]. The exact binding site of 10H10 mAb to TBEV and other flaviviruses is not known. Understanding the mechanisms of 10H10 mAb cross-reactivity with different viruses can help to develop therapeutic antibodies and improve vaccination strategies for different flaviviruses. Using experimental and bioinformatic approaches, we identified the sites of TEBV and ZIKV E-protein recognition by 10H10.

## 2. Results and Discussion

### 2.1. 10H10 Antibody Epitope Mapping

It is known that the major targets of the humoral immune response against flaviviruses are the surface E protein and NS1 [17,18]. It was shown previously that the 10H10 antibody binds to fragments of the E protein of WNV at 1–180 and 53–126 aa. It was also shown that 10H10 mAb does not bind part of the WNV E protein at 1–86 aa [28]. These data suggest that the possible site recognized by the 10H10 antibody is located between 86 and 126 aa of the WNV E protein. However, the exact E protein epitope recognized by 10H10 mAb is not known. To map the flavivirus polyprotein region which is recognized by 10H10 mAb, recombinant proteins including various domains of the E protein (TEF1, TEF2 и TEF3) and NS1 (TNS1) protein of TBEV were constructed (Figure 1). In addition, recombinant proteins consisting of ZIKV, DENV and WNV, ZEF1, DEF1 and WEF1, respectively, able to identify TEF1 were produced (Figure 1). The identity of TBEV, ZIKV, DENV and WNV flavivirus E proteins are shown in Appendix A. To increase the stability and yield of the recombinant protein, a thioredoxin sequence was added to the N-terminus of each construct [29]. The proteins were expressed in *E. coli* and purified using metal-chelate affinity chromatography as described in the Materials and Methods section. The generated recombinant proteins used in the experiments are listed in Figure 1.

The results of dot-blot analysis and ELISA of 10H10 antibody interaction with the above recombinant proteins are shown in Figure 2. The data presented in Figure 2a indicate that 10H10 mAb binds TEF1, WEF1, ZEF1 and DEF1 proteins but does not bind TEF2, TEF3 or TNS1. The binding of 10H10 mAb to TEF1, WEF1, ZEF1 and DEF1 was also confirmed by ELISA (Figure 2b). It is important to note that only E fragments of polyproteins consisting of 180 aa interact with 10H10 mAb. This observation confirms previous findings that the site recognized by 10H10 is located between 86 and 126 aa of the E protein [28]. The observed difference in the signal between the West Nile virus fragment and others used in the work can be explained by the difference in the amino acid composition of the regions flanking the fusion loop. As a result, amino acid residues can contribute to the strength of the interaction of proteins with the 10H10 antibody.

The above data indicate that 10H10 mAb is a cross-reactive antibody since it can recognize sites within the first 180 aa of the E protein from different flavivirus spices. The alignment of the amino acid sequences of the E protein from TBEV, JEV, WNV, YFV, ZIKV and DENV (I-IV) (Appendix A) shows that only two regions have an identity. One highly identified region is located within the stem (S) part of the E protein (Figure 3) but is not recognized by the 10H10 antibody, as shown by dot-blot analysis (Figure 2a TEF3). The second region with high identification between different flaviviruses is represented by the fusion loop of E protein domain II (from 95 to 115 aa) (Figure 3b).

The amino acid alignment in Figure 3b shows that the fusion loop of the E protein (amino acid residues 98-DRGWGNGCGLFGKGSL-113) is the most conserved among the different flavivirus species. Based on the above data, we hypothesized that the region with the highest identity harboring amino acid residues 98-DRGWGNGCGLFGKGSL-113 of different flavivirus species (Figure 3b) is the target for of cross-reactive 10H10 antibody. To verify this hypothesis, we created a 3D model of the 10H10 antibody binding part and 3D model for the complex of the 10H10 antibody interacting with the FL part of ZIKV and TBEV E proteins.

### 2.2. The Structure of 10H10 Antibody Variable Domains

Nucleotide sequences of the 10H10 antibody’s VH and VL regions were obtained from 10H10 hybridoma total RNA as described in the Materials and Methods section. The amino acid sequences were translated from the above nucleotide sequences deposited in the NCBI GenBank under the accession numbers OK483332 (VH) and OL448869 (VL). The amino acid sequence is presented in the Appendix A. The 3D model of 10H10 antibody variable domains was created using the deposited 10H10 amino acid sequences with SAbPred tools [30]. The algorithm imbedded the SAbPred tools’ selected VH and VL regions available in the database sequences with 80 ± 2.5% identity in the 10H10 VH and VL domains. The 3D structure of the 10H10 VH and VL domains was created based on the existing VH/VL template in the SAbPred database. Using the template with 80% sequence identity to target, we obtained a confidence interval of 75% that the VH and VL framework was modelled to within 1Å RMSD. The created 3D model of 10H10 variable domains (Figure 4) was used to analyze its binding to the FL part of ZIKV and TBEV E proteins.

A tertiary structure evaluation of the final assembled protein was performed followed by protein-structure refinement using the Schrodinger Suite: Release 2021-1 [31] software package program where the Ramachandran plot statistics indicated a good quality for structures of VH and VL antibody 10H10. More than 80% of their residues had proper φ and ψ torsion angles which were placed in the permitted areas. Indeed, 80% of amino acids of 10H10 were located in the most favored and permitted regions (Appendix A). Protein models are of a good quality when over 90% of their residues are located in the most favored regions [32].

### 2.3. Molecular Modeling of 10H10 Variable Domains Binding to E Protein

To analyze the interaction between VH-VL of the 10H10 antibody and E protein, molecular protein–protein docking was performed. Based on the binding of 10H10 mAb to the region within the first 180 amino acids of the E protein (Figure 2a) and the most notable identity of flavivirus E proteins between cysteines at 92 and 116 position (Figure 3b, Appendix A), the fusion loop was selected as a putative binding site to perform a molecular docking simulation (Figure 5). The cryo-EM structure of ZIKV (5H37) and the crystallographically determined structure of the soluble E protein from TBEV (1SVB) were downloaded from Protein Data Bank [www.rcsb.org] (accessed on 23 March 2022). In both cases, only one polypeptide chain (chain A) corresponding to the E protein was chosen for molecular modelling. During molecular protein docking, several calculations corresponding to different position of the 10H10 VH/VL complex 3D model (Figure 4) relative to the E protein of TBEV and ZIKV were obtained. The most energetically favorable position of VH-VL chains of the 10H10 mAb was located within the fusion loop of both TBEV and ZIKV E proteins (Figure 5a,b). Therefore, the 3D model confirmed our hypothesis that the binding site of 10H10 mAb is located within the FL domain of the E protein.

Further analysis revealed that the interaction of 10H10 VH/VL with the TBEV and ZIKV the FL part of E protein is characterized by several intermolecular interactions. The TBEV-10H10 complex (Figure 5d, Appendix A) is formed by hydrogen bridges between the amino acids of the fusion loop (marked as A) and the following antibodies 10H10 VH/VL (marked as H and L, respectively): D_A_98 and H_H_38, G_A_102 and Y_H_40, H_A_104 and D_L_31. Stacking π-π bonds were observed between the aromatic rings of H_A_104 and F_L_114. In addition, a hydrogen bond formed between T_A_76 and K_L_38 that was further away from the fusion loop of the E protein.

The ZIKV-10H10 complex (Figure 5e) has more potential intermolecular interactions compared to the TBEV-10H10 complex. Hydrogen binds formed between the following amino acids: R_A_99 and Y_H_40, R_A_99 and Q_L_116, G_A_104 and R_L_52, C_A_105 and R_L_52; and salt bridges between D_A_98 and K_L_38, K_A_110 and D_H_98. Stacking π-π interactions formed between F_A_108 and Y_H_40. Additionally, hydrogen binds formed between K_A_251 and Y_L_38, S_A_72 and K_L_36; and salt bridges between K_A_251 and D_L_31, K_A_251 and D_L_34 that do not belong to the FL of the E protein. Therefore, there are common and individual intramolecular interactions between 10H10 mAb and E proteins of Zika and tick-borne encephalitis viruses. It is conceivable that the strongest and most important interaction between the 10H10 antibody and the flavivirus E proteins occurs within the space between 98 and 105 aa. It appears that the binding site of 10H10 mAb to flaviviruses’ E proteins is a conformational epitope that allows for cross-reactivity and relies on several intermolecular interaction mechanisms that include hydrogen bonds, salt bridges and stacking π-π interactions.

Protein–protein complexes of TBEV-10H10 and ZIKV-10H10 presented in Figure 5a,b were used for subsequent molecular dynamics simulations. The structural integrity of the complexes was preserved throughout the calculation. The formation of hydrogen bonds between the FL of flaviviruses E proteins and 10H10 antibody chains were recorded (Figure 6a). The number of hydrogen bonds and other molecular interactions reflects the basic parameter of protein–protein interaction stability.

Up to nine hydrogen bonds were registered during the entire simulation time between the amino acids of TBEV E protein FL domain and the 10H10 antibody (depicted on Figure 6a). At around 40,000 ps (40 ns), a slight perturbation of the RMSD (Appendix A) value was observed, accompanied by an increase in the number of hydrogen bonds (Figure 6a). The increase in the H-bonds number and other intramolecular contacts were recorded compared to the initial simulation (Figure 5d). Hydrogen bonds were observed between the following amino acids pairs: R_A_73-D_H_58; C_A_74-D_H_58; T_A_76-Y_H_38; H_A_108-Y_H_38; R_A_99-D_H_36; G_A_106-Q_L_116. π-π stacking interactions occurred between the following pairs: H_A_104-Y_H_37 and W_A_101-W_L_105 (Figure 6b). The rearrangement of proteins relative to each other during the simulation was observed when compared to the starting position (Figure 5d). The system equilibrated after 50,000 (50 ns) ps of simulation, while the RMSD fluctuation did not exceed 4 Å. The position of proteins relative to each other noticeably changed as early as 40,000 ps of simulation time. The number of registered hydrogen bonds decreased. By the end of the simulations, hydrogen bonds were registered for the following pairs: A_A_72-Y_H_38; R_A_99-Y_H_37; H_A_104–Y_H_38; salt-bridges for R_A_73-D_H_36; R_A_98-K_L_36; and π-π stacking for H_A_104-Y_H_40. Only one hydrogen bridge between H_A_104 and Y_H_37 was preserved during the entirety of TBEV-10H10 complex calculation (Figure 6b,c).

The ZIKV-10H10 complex is characterized by more intermolecular interactions compared to the TBEV-10H10 complex. On average, about 11 interactions were registered in the ZIKV-10H10 complex (Figure 6d and Appendix A) compared to 4–5 interactions in TBEV-10H10 complex (Figure 6a). The intermolecular interactions within the ZIKV-10H10 complex consisted of hydrogen bonds in the following amino acid pairs: D_A_98-K_L_36; C_A_105-R_L_52; K_A_251-D_L_31; K_A_251-Y_L_38, and salt bridges in D_A_98-K_L_36; K_A_251-D_L_31 and K_A_251-D_L_34 remained stable during the entire simulation (Figure 6e,f and Appendix A). The data suggest that the ZIKV-10H10 complex forms a stable structure and the position of proteins does not change relative to each other during the molecular dynamic simulation.

The results of molecular modeling indicate that 10H10 mAb can form stable complexes with the FL region of TBEV and ZIKV E proteins that involve various intermolecular interactions including hydrogen bonds, salt bridges and π-π stacking. The interaction of TBEV FL with 10H10 mAb is likely more flexible as suggested by fewer H-bonds registered during the simulation, while the recognition of the FL region of the ZIKV E protein by 10H10 mAb is provided by more H-bonds. Despite the fact that 10H10 mAbs were generated by immunizing mice with TBEV Ag [27], it is conceivable that 10H10 mAb has a higher affinity to the FL region of the ZIKV E protein as compared to TBEV FL.

Our results indicate that the fusion loop may act as a target for the 10H10 antibody, and this may explain the broad reactivity to various flaviviruses observed for it. It can be argued that the interaction with amino acid residues of the fusion loop makes the greatest contribution to the energy of the complex, while other residues can also play a role in this interaction, increasing or decreasing the total value. Our simulations suggest a possible geometry and energy, but more appropriate experimental methods, such as an X-ray diffraction analysis, may reveal the exact structure of the fusion loop-antibody complex.

## 3. Materials and Methods

### 3.1. Construction of Recombinant Plasmids

The DNA constructs were designed based on the complete genome sequences of four flaviviruses, namely ZIKV (KX377335.1), TBEV (U39292.1), DENV 3 (AY099336.1) and WNV (NC_001563.2)). The nucleotide sequences encoding fragments of corresponding flavivirus polyproteins were codon-optimized and synthesized by DNA-Synthesis ltd (Russia). DNA encoding TEF1, ZEF1, WEF1, DEF1, TEF2, TEF3 and TNS1 fragments (see Figure 1) were amplified by PCR using Forward and Reverse primers (Appendix A), digested with NcoI and XhoI (TBEV and ZIKV), BamHI and XhoI (DENV and WNV) restriction sites, and cloned into a pET32 expression vector. For subsequent purification of the protein, nucleotide sequences encoding fragments of corresponding flavivirus polyproteins were cloned in frame with the C-terminal His_6_ tag. The nucleotide sequences of recombinant plasmids were verified by Sanger sequencing.

### 3.2. E. coli Transformation and Recombinant Proteins Production

For the production of recombinant proteins, *E. coli* strain BL21(DE3) was chemically transformed. Individual *E. coli* colonies containing recombinant plasmids were selected and cultured overnight on an orbital shaker (Biosan, Riga, Latvia) at 37 °C 180 rpm in LB medium containing 100 μg/mL ampicillin. A total of 1 mM IPTG (Anatrace Products, Maumee USA) was added to the mixture after the optical density (OD600) reached 0.8 and was cultivated at 18 °C for 16 h. Cells were centrifuged at 6000× *g* for 15 min at 4 °C, lysed with lysis buffer (30 mM NaH_2_PO_4_, 0.5 M NaCl, 20 mM imidazole, 8 M Urea, 0.1% Triton X-100, pH 7.4) and sonicated. The cell lysates were cleared from debris by centrifugation at 15,000× *g* for 25 min at 4 °C. Recombinant proteins were purified from cleared lysates using metal chelate affinity chromatography on Ni-IMAC Sepharose sorbent (GE Healthcare, Chicago, IL, USA). Fractions were collected in 15 mL. The obtained proteins’ quality was checked using electrophoresis under denaturing conditions in a 15% polyacrylamide gel according to the Laemmli method [33].

### 3.3. Dot-Blot

Recombinant proteins at a concentration of 1.5 mg/mL in 1 µL were adsorbed onto a nitrocellulose membrane. Next, the membranes were cultured in a blocking buffer containing PBS, 0.1% Tween-20 with 1% BSA. After washing three times with PBS + 0.1%, Tween-20 10H10 mAb 2 μg/mL in blocking buffer was added and incubated for 10 min. After incubation, membranes were washed three times and incubated for 10 min with goat anti-Mouse IgG polyclonal antibody 0.25 μg/mL conjugated with alkaline phosphatase (BioRad, Berkeley, CA, USA). After washing 3 times, membranes were incubated for 5 min with an aqueous solution of 5-bromo-4-chloro-3-indolyl phosphate (BCIP) and nitrosinetetrazole (NTB) (Invitrogen, Waltham, MA, USA). The enzymatic reaction was stopped by washing with distilled water.

### 3.4. ELISA

Nunc high-sorption strip plates (Thermo Scientific, Waltham, MA, USA) were coated with 200 ng/well/100 μL recombinant proteins in Tris buffer (0.15 M NaCl; 0.02 M Tris-HCl, pH 7.4, TBS) at 4 °C for 18 h. Coated plates were rinsed three times with PBS containing 0.1% Tween 20 (PBST) and blocked with 1% casein in PBS. The 10H10 antibody 3 μg/mL in blocking buffer in various dilutions (as shown in Figure 2) was added to the wells and incubated for 1 h at 37 °C. After washing 3 times with PBST, the plates were incubated for 1 h at 37 °C with goat anti-Mouse IgG 0.5 μg/mL (Invitrogen, Waltham, MA, USA). Plates were washed three times with PBST. TMB substrate (Termo Scientific, Waltham, MA, USA) was added to the wells and plates were incubated for 10 min. The enzymatic reaction was stopped with 1 N HCl. The optical density (OD) was measured on a VarioScan 6Lux (Thermo Scientific, Waltham, MA, USA) spectrophotometer at a wavelength of 450 nm.

### 3.5. Alignment of Amino Acid Sequences

Alignment of the amino acid sequences of the E protein of the following nine viruses: tick-borne encephalitis virus-strain Hypr (U39292.1), Japanese encephalitis virus-strain Kamiyama 1 (S47265.1), yellow fever virus-strain 17D Tiantan (FJ654700.1), West Nile virus-strain 956 (NC_001563.2), Zika virus-strain MR-766 (KX377335.1), dengue virus 1—strain Nauru/West Pac/1974 (U88536.1), dengue virus 2—strain 16,681 (M84728.1)), dengue virus 3—strain Sri Lanka/1266/2000 (AY099336.1), dengue virus 4—strain Dominica 814,669 (AF326825.1) using the BioEdit program with the Clustal W Multiple Alignment method was performed.

### 3.6. Sequencing of the VH and VL Sequence of the 10H10 Antibody

Total RNA was isolated from 10H10 mouse hybridoma cells [26] and reversed into cDNA using Maxima Reverse Transcriptase (Thermo Scientific, Waltham, MA, USA). The reaction was carried out for 30 min at 50 °C according to the manufacturer’s recommendations, using 5 µg of total RNA and 20 pmol of gene-specific reverse primers (AbHC_mIgG2B_R, AbLC_3XKc_R) (Appendix A) [34].

Amplification of the cDNA fragment was carried out using the above reverse primers paired with the corresponding forward primers—for the heavy chain with six degenerate nucleotide positions (AbHC_MH1_F, Biosset, Novosibirsk, Russia) and for the light chain with seven degenerate nucleotide positions (AbLC_5XMk_F, Biosset, Novosibirsk, Russia). PCR conditions were selected using the primer annealing temperature gradient of the total cDNA template. The reaction mixture contained 1x Phusion HF buffer, 0.2 mM dNTP, 2.0 mM free Mg^2+^, 0.4 µM forward and reverse primers (AbHC_MH1_F paired with AbHC_mIgG2B_R or AbLC_5XMk_F and AbLC_3XKc_R) (Appendix A), an aliquot of the RT reaction, and 10 u/mL Phusion Hot Start II DNA polymerase. The optimal temperature profile of amplification according to the results of gradient PCR consisted of incubation at 98 °C for 1 min, and then 35 cycles, consisting of the stage of incubation at 98 °C for 10 s, annealing at 60 °C for 15 s and elongation at 72 °C for 20 s, with a final incubation step at 72 °C for 7 min and storage at 4 °C.

The resulting PCR products of about 400 bp in length were cloned using CloneJET PCR Cloning Kit (Thermo Scientific, Waltham, MA, USA). Several clones for each PCR fragment were sequenced with the Sanger method using an ABI PRIZM 3130XL automated gene analyzer (Applied Biosystems, USA).

A typical Sanger reaction in a volume of 40 µL contained the following: 300 fmol of plasmid DNA, 20 pmol of a vector-specific primer (JET_F or JET_R), 1X sequencing buffer, and 2 µL of BigDye v.3.1 reagent (Applied Biosystems, USA). The reaction temperature profile consisted of initial melting at 95 °C for 2 min followed by 50 cycles of incubation at 95 °C for 25 s, annealing at 50 °C for 5 s and elongation at 60 °C for 4 min. Sanger reactions were purified from unincorporated fluorescently labeled ddNTPs by centrifugation through individual columns with Sephadex G-50 Fine gel filtration media (900× *g*, 2 min). Sequencing traces were assembled into a contig by academic version of Vector NTI 10 (Invitrogene, Waltham, MA, USA) (Appendix A).The resulting gene sequences encoding VH and VL antibodies 10H10 were deposited in the NCBI GenBank database (https://www.ncbi.nlm.nih.gov (accessed on 23 March 2022)) under accession numbers OK483332 and OL448869, respectively.

### 3.7. Purification of the 10H10 Monoclonal Antibody Preparation

Ascitic fluid containing 10H10 antibodies were kindly provided by Prof. V.Loktev. Ascitic fluid was diluted in 20 mL PBS, and centrifuged at 12,000× *g* for 25 min at 4 °C. Purification of monoclonal antibodies was performed using protein A affinity chromatography with MabSelect SURE sorbent (GE Healthcare). Fractions were collected in 15 mL. Antibody quality was controlled by electrophoresis under denaturing conditions in a 12% polyacrylamide gel according to the Laemmli method [33].

### 3.8. Structure of 10H10

The structure of the antibody (the amino acid sequence is presented in the Appendix A) was modeled using the SAbPred tool, which allows for the automatic reproduction of the probable structure of variable antibody fragments using the ABodyBuilder algorithm. An antibody model was built by recognizing the framework regions and CDRs of the amino acid sequence using ANARCI. Examples of light and heavy chain variable fragment framework regions were selected by SAbDab and the choice of orientation relative to each other by ABangle. FREAD was used to predict CDR conformation. If the conformation could not be predicted, the MODELER tool was used, which models the CDR cycle. SCRWL4 was used to predict the confirmation of side chains, which were calculated based on the amino acid sequence [30].

### 3.9. Protein Docking the Structure of Antigens and Antibodies

#### 3.9.1. Protein Preparation

Crystallographic structures of the glycoprotein ZIKV (PDB code 5H37 [10.1038/ncomms13679]) and TBEV (PBD code 1SVB [nature.com/articles/375291a0]) were downloaded from the non-commercial Protein Data Bank [10.1093/nar/28.1.235]. Model structures of glycoproteins and 10H10 antibody were prepared for calculation using Schrodinger Protein prepwizard tools. Hydrogen atoms were added and minimized, missing amino acid side chains were added, bond multiplicities were restored, and solvent molecules were removed. One chain was chosen in the glycoprotein. Protein structures were limitedly optimized in the OPLS3e force field [10.1021/ct300203w] at physiological pH values.

#### 3.9.2. Molecular Docking Procedure

The molecular protein docking procedure was performed using the PIPER plugin [10.1002/prot.21117] implemented in the Schrodinger Suite: Release 2021-1 software package [31]. The main algorithm of the plugin is to rotate the ligand (with a change in conformation if necessary) relative to the receptor to find the optimal scoring function value (piper score) of the ligand-receptor position. The selection of docking results was carried out by evaluating the piper energy—the energy of unbound interactions between the ligand and the receptor in the zone of active contact. The FL in the glycoprotein containing the following amino acid sequence: 92-CKRTLVDRGWGNGCGLFGKGSLVTC-116, was chosen as the active site. Glycoproteins ZIKV (Z) and TBEV (T) were considered as receptors and the 10H10 antibody as a ligand.

#### 3.9.3. Molecular Dynamics

To build a molecular dynamics simulation model, the most energetically favorable positions of the systems receptor (glycoproteins Z and T)—ligand (antibody 10H10) were chosen.

Peptide–protein complexes Z-10H10 and T-10H10 were placed in an orthorhombic system, with a buffer zone of 15 Å from the protein surface. The system was filled with 0.15 M NaCl aqueous solution. The solvent model was TIP3P. Environment—NVT. The period of the recorded dynamics simulation was 100 nanoseconds at a temperature of 310 K (37 °C). The protocol for preparing the system for simulation included preliminary minimization and balancing of the system components. The calculations were carried out using the Desmond program [31] implemented in the Schrodinger Suite: Release 2021-1.

## 4. Conclusions

10H10 mAb can recognize the FL region of E proteins from different flaviviruses which depends on multimolecular interactions. The binding site of 10H10mAb is likely located within the fusion loop of the flaviviruses’ E proteins consisting of 98-DRGWGNGCGLFGKGSL-113 amino acids. Overall, the results indicate that the antibody-antigen complex can form a rigid or dynamic structure that provides efficient protein–protein interactions and antibody cross reactivity.

## Figures and Tables

**Figure 1 ijms-23-07721-f001:**
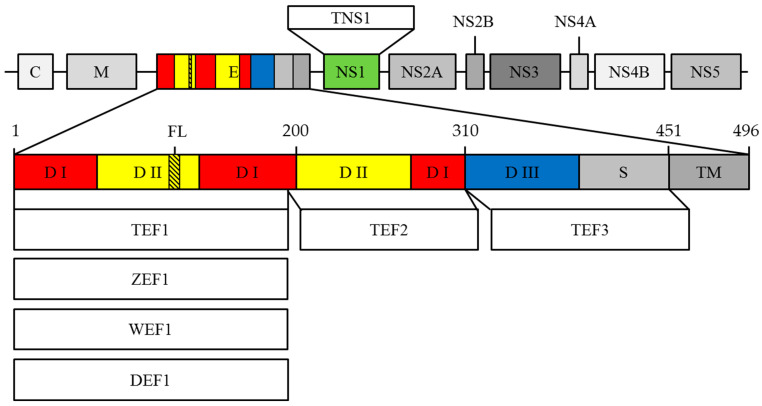
Schematic visualization of fragments polyprotein structure. C—capsid protein; M—membrane protein; E—envelope protein; NS1-NS5—seven non-structural proteins; DI—domain I; DII—domain II; DIII—domain III; S—stem; TM—transmembrane region; FL—fusion loop; TEF1—the fragment of TBEV E protein containing 1–187 aa long; ZEF1—the fragment of ZIKV E protein containing 1–187 aa long; WEF1—the fragment of WNV E protein containing 1–186 aa long; DEF1—the fragment of DENV E protein containing 1–184 aa long; TEF2—the fragment of TBEV E protein containing 188–310 aa long; TNS1—NS1 protein of the TBEV; TEF3—the fragment of TBEV E protein containing 311–451 aa long.

**Figure 2 ijms-23-07721-f002:**
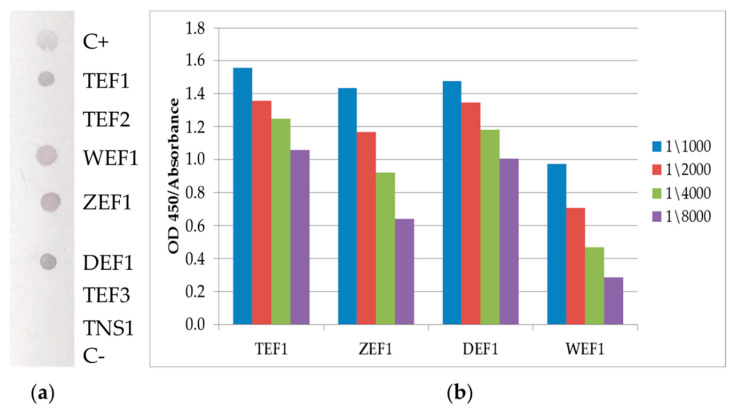
The interaction of the antibody with fragments of viral proteins. (**a**) Dot blot analysis: The membrane was blotted with C+—mouse antibody (positive control); Fragments polyproteins TEF1, TEF2, WEF1, ZEF1, DEF1, TEF3, TNS1 (as shown), C−—RBD SARS-CoV-2 (negative control). After that, the membrane was incubated with 10H10 mAb. Binding of 10H10 to immobilized proteins was detected with secondary anti-mouse antibody conjugated with AP. (**b**) ELISA: TEF1, WEF1, DEF1 and ZEF1 proteins were immobilized on 96-well plates. 10H10 mAb were added to wells. Binding of 10H10 to immobilized protein was detected with secondary anti-mouse antibody conjugated with HRP.

**Figure 3 ijms-23-07721-f003:**
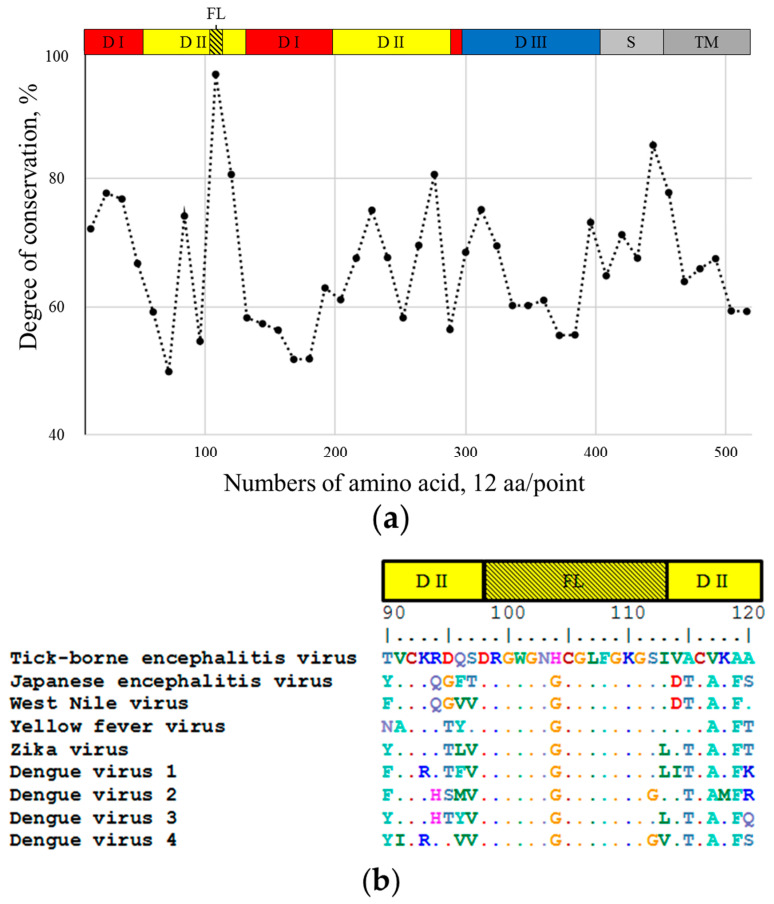
The identity of E proteins from 9 different flaviviruses. DI—domain I; DII—domain II; DIII—domain III; S—stem; TM—transmembrane region; FL—fusion loop. (**a**)—for better perception, the sequence was divided into equal segments containing 12 aa each and the degree of conversation was calculated. The X axis displays amino acid residues of E protein, the Y axis reflects the degree of conservation of the amino acid residue in different flaviviruses (Tick-borne encephalitis virus-strain Hypr (U39292.1), Japanese encephalitis virus-strain Kamiyama 1 (S47265.1), Yellow fever virus-strain 17D Tiantan (FJ654700.1), West Nile virus-strain 956 (NC_001563.2), Zika virus-strain MR-766 (KX377335.1), Dengue virus 1-4-strain Nauru/West Pac/1974 (U88536.1), strain 16,681 (M84728.1), strain Sri Lanka/1266/2000 (AY099336.1), strain Dominica 814,669 (AF326825.1)). (**b**)—amino acid sequence alignment of the indicated viruses FL regions.

**Figure 4 ijms-23-07721-f004:**
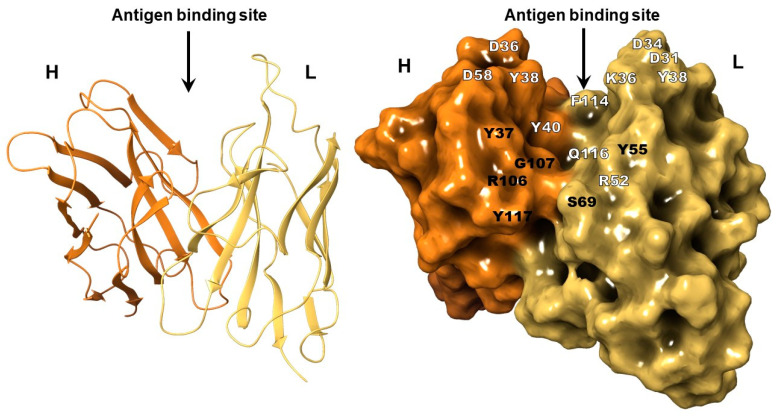
3D model of the VH and VL complex of the 10H10 antibody: amino acids reactive with FL region of E protein are marked white color (hereinafter, H and L are name of chain).

**Figure 5 ijms-23-07721-f005:**
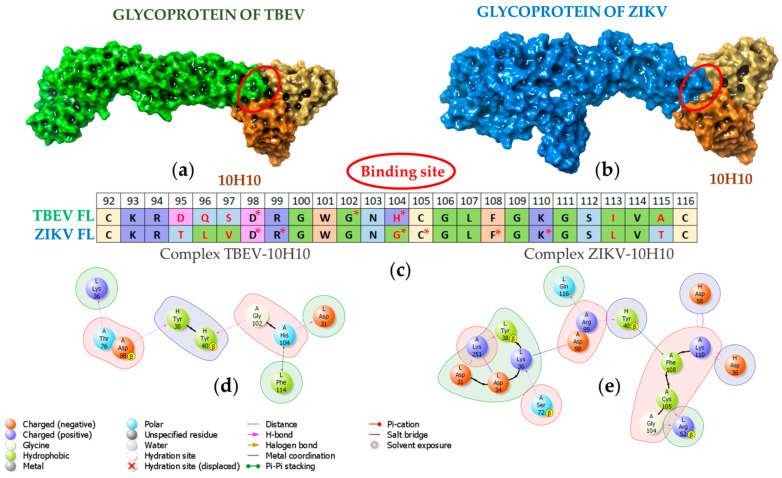
Visualization of flavivirus glycoproteins and 10H10 antibody protein–protein docking. (**a**) Docking position of the interaction of TBEV surface glycoprotein (E-protein) and 10H10 antibody (TBEV-10H10); (**b**) Docking position of the interaction between ZIKV E protein and 10H10 antibody (ZIKV-10H10); (**c**) Amino acids of the fusion loop included in the active binding site. Amino acids forming intramolecular interactions between VH/VL and FL are marked with asterisk; (**d**,**e**) The main intermolecular interactions between the amino acids of the FL and the antibody. Chain A is an amino acid sequence of the FL of TBEV and ZIKV; chains H and L correspond to VH and VL of 10H10 antibody.

**Figure 6 ijms-23-07721-f006:**
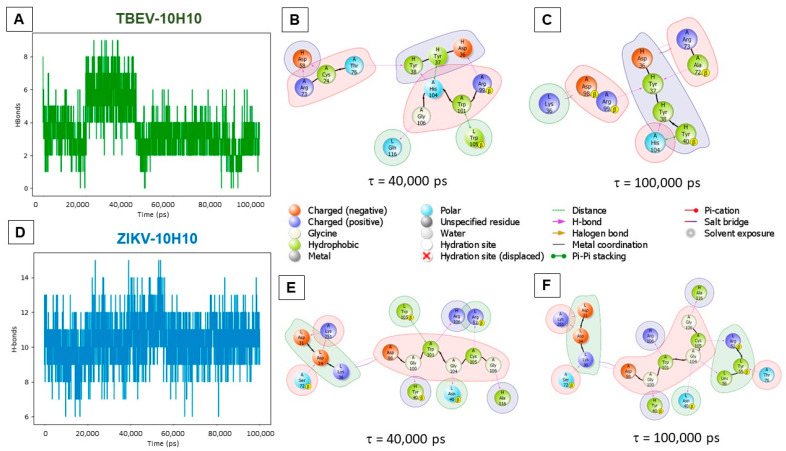
Visualization of molecular-dynamics simulation: (**A**,**D**)—the total number of hydrogen bonds between the amino acids of glycoproteins and the 10H10 antibody recorded during the entire simulation time (the spheres show changes in the secondary structure of the FL (green) and of the antibody); (**B**,**C**,**E**,**F**)—visualization of intermolecular interactions in TBEV-10H10 and ZIKV-10H10 complexes.

## Data Availability

Not applicable.

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
