# Peer review of "Can Modern Molecular Modeling Methods Help Find the Area of Potential Vulnerability of Flaviviruses?"

_ijms, 2022, doi:10.3390/ijms23147721_

Round 1

Reviewer 1 Report

This study is a well-done examination of the binding properties of 10H10, with results indicating an epitope of the fusion loop of Flaviviruses, including predictions of increased ZIKV binding over TBEV.  This adds to the body of work that is present regarding the cross-reactivity of Flaviviruses and may have implications for immunotherapy and vaccination.  My comments are minor.

Though understandable, the English needs to be more thoroughly edited for accuracy.

Addition of some text indicating how this may translate to neutralization capacities with various Flavivirus species is indicated here.  Especially as it relates to the potential for antibody dependent enhancement of heterologous species.

Under the plasmid construction section, it would be helpful to add the tag you used for purification.  I assume you cloned in frame for the C-terminal His6 tag in your vector due to your use of Ni for purification, but it should be spelled out.

Did you refold the proteins?  If so, you need to add how they were refolded.

What controls were used for the ELISA?  Was the data in Figure 2 background subtracted?

Author Response

  1. Addition of some text indicating how this may translate to neutralization capacities with various Flavivirus species is indicated here. Especially as it relates to the potential for antibody dependent enhancement of heterologous species.

Response:

Thank you for your comment. The 10H10 antibody is non-neutralizing.

We added the following sentence to the text:

Previously we have determined and described that 10H10 antibody interacting with TBEV and other flaviviruses is non-neutralizing Ab [27].

  1. Under the plasmid construction section, it would be helpful to add the tag you used for purification.  I assume you cloned in frame for the C-terminal His6tag in your vector due to your use of Ni for purification, but it should be spelled out.

Response:

Thank you for pointing on missing information for tag and protein purification. We have added protein purification description to the Materials and Methods section.

The following sentence was added to the manuscript:

For subsequent purification of the protein a nucleotide sequences encoding fragments of corresponding flavivirus polyproteins were cloned in frame with C-terminal HIS tag.

  1. Did you refold the proteins? If so, you need to add how they were refolded.

Response:

The proteins were not refolded.

What controls were used for the ELISA? Was the data in Figure 2 background subtracted?

Response:

Thank you for your comment. To measure the background signal, the ELISA procedure included a well without antigen and a well with an protein antigen (negative control) that does not interact with the 10H10 antibody. The wells containing antigen must have an absorbance that is at least twice the absorbance of the background wells for the results to be interpretable. This allows us to speak about the correctness of the study and the absence of a false positive result due to the high background. As a result of the study, the background value was subtracted from the OD values for each interaction of the antibody with the antigen. Due to the fact that the negative control showed comparable values with the background values, the value of the negative control is 0 and therefore is not given.

Reviewer 2 Report

Major comments:

1. In the optical density measurement as shown in Fig. 2b, the values reflect the binding affinity of 10H10 mAb to the fragment polyproteins (TEF1, ZEF1, DEF1, and WEF1), and for example, the OD450 of WEF1 looks smaller than the other fragments. I am very curious whether it is possible to explain the difference in the absorbance values between the fragments with the differences in the amino acid sequences of the FL loop. If possible, please discuss this.

2. The molecular docking study of 10H10 mAb and TEBV or ZIKV E-proteins showed that the usage of amino acid residues of the E-proteins in contact with those of antibody was different. This indicates that there are at least two binding modes between 10H10 mAb and E-proteins even though the amino acid sequences in the FL-loop were highly conserved among the viruses. What does originate from the different binding modes between the two antibody-E-protein complexes? Is it possible to explain from the MD simulations?

3. For homology modeling, the MODELLER software, which is one the most popular software was used, however, there are problems with accuracy. If you use the state-of-the-art method such as AlphaFold2, how the model structure would be changed?

Minor:

1.     In Fig.5, what is glycoprotein? Is it the same as E-protein?

2.   Page 9 Line 273 for example, it is common that the time unit of MD simulation is represented as nano-second (ns) instead of pico-second (ps).

Typos:

1.     Page 8 Line 252, hydrogen bonds?

2.     Page 7 Line 232, revealed?

3.     Page 9 Line 292, intermolecular interactions?   

Author Response

  1. In the optical density measurement as shown in Fig. 2b, the values reflect the binding affinity of 10H10 mAb to the fragment polyproteins (TEF1, ZEF1, DEF1, and WEF1), and for example, the OD450 of WEF1 looks smaller than the other fragments. I am very curious whether it is possible to explain the difference in the absorbance values between the fragments with the differences in the amino acid sequences of the FL loop. If possible, please discuss this.

Response:

We included to the Discussion section a sentence to explain the difference in absorbance values between fragments with the different amino acid sequences of the FL loop.

Addition to the manuscript:

The difference in optical density between the West Nile virus fragment (WEF1) and other proteins interacting with 10H10 Ab can be explained by the amino acid composition of FL loop. Certain amino acid residues can contribute to the strength of FL segment interaction with 10H10 antibody.

  1. The molecular docking study of 10H10 mAb and TEBV or ZIKV E-proteins showed that the usage of amino acid residues of the E-proteins in contact with those of antibody was different. This indicates that there are at least two binding modes between 10H10 mAb and E-proteins even though the amino acid sequences in the FL-loop were highly conserved among the viruses. What does originate from the different binding modes between the two antibody-E-protein complexes? Is it possible to explain from the MD simulations?

Response:

We do not agree that we have two binding modes. The fusion loop is a highly mobile part of the protein. The key area of interaction of the antibody with the fusion loop in both cases is amino acids 98 to 110. Of course, we observe a difference in interaction with other amino acids of the fusion loop. The reason lies in the mobility of the loop itself. Based on the results of molecular modeling, we can note that the antibody with the Zika virus fusion loop forms more intermolecular interactions than with the tick-borne encephalitis virus loop observed at certain simulation times. But this does not mean at all that the affinity of the antibody to the fusion loop of the tick-borne encephalitis virus is less pronounced.

  1. For homology modeling, the MODELLER software, which is one the most popular software was used, however, there are problems with accuracy. If you use the state-of-the-art method such as AlphaFold2, how the model structure would be changed?

Response: Thank you for suggestion to use different software. In our report, we have achieved RMSD value within 1 Å. In addition the Ramachandran map presented in supplementary materials indicates that almost all amino acid dihedral angles are located in the optimal energy region for 10H10 Ab and FL interaction. We believe that the result of folding in AlpfaFold2 will not critically affect the main conclusions of the paper.

Minor comments:

  1. In Fig.5, what is glycoprotein? Is it the same as E-protein?

Surface glycoprotein and E-protein are the same. The text has been corrected.

  1. Page 9 Line 273 for example, it is common that the time unit of MD simulation is represented as nano-second (ns) instead of pico-second (ps).

We have corrected the values in the text.

Typos:

  1. Page 8 Line 252, hydrogen bonds?

The text has been corrected.

  1. Page 7 Line 232, revealed?

The text has been corrected.

  1. Page 9 Line 292, intermolecular interactions?

The text has been corrected.